The role of multiple negative social relationships in inflammatory cytokine responses to a laboratory stressor

Song Sunmi 1
Graham-Engeland Jennifer E. 2
Corwin Elizabeth J. 3
Ceballos Rachel M. 4
Taylor Shelley E. 5
Seeman Teresa 6
Klein Laura Cousino 7 lcklein@psu.edu
1 Social Behavioral Research Branch, National Human Genome Research Institute, National Institutes of Health , USA
2 Department of Biobehavioral Health, The Pennsylvania State University , USA
3 Nell Hodgson Woodruff School of Nursing, Emory University , USA
4 Cancer Prevention Program, Fred Hutchinson Cancer Research Center , USA
5 Department of Psychology, University of California , Los Angeles , USA
6 Division of Geriatrics, David Geffen School of Medicine, University of California , Los Angeles , USA
7 Department of Biobehavioral Health and Penn State Institute of the Neuroscience, The Pennsylvania State University , USA
D’Acquisto Fulvio
Electronic publication date: 2015 Jun 2
Publication date: 2015
Volume: 3
Electronic Location ID: e959
Received 2015 Jan 20; Accepted 2015 Apr 24
Copyright: © 2015 Song et al.
Copyright year: 2015
Copyright holder: Song et al.
License: This is an open access article distributed under the terms of the Creative Commons Attribution License, which permits unrestricted use, distribution, reproduction and adaptation in any medium and for any purpose provided that it is properly attributed. For attribution, the original author(s), title, publication source (PeerJ) and either DOI or URL of the article must be cited.
License URL: https://creativecommons.org/licenses/by/4.0/

Keywords: Stress, Multiple negative social relationships, Hostility, Inflammatory cytokine response, Depressed mood, A Trier Social Stress Task

Funding: General Clinical Research Center of The Pennsylvania State University MO1-RR-10732 National Science Foundataion SBR9905157 College of Health and Human Development of The Pennsylvania State University 223-15-3605 Kligman Graduate Fellowship This research was conducted at The Pennsylvania State University. Funding was provided by the General Clinical Research Center of The Pennsylvania State University (NIH grant MO1-RR-10732). Funding was also provided by the National Science Foundataion (SBR9905157; SET and LCK) and a grant from the College of Health and Human Development of The Pennsylvania State University (223-15-3605; LCK and EJC). The first author is currently a postdoctoral fellow at the Social and Behavioral Research Branch at the National Human Genome Research Institute and was supported by a Kligman Graduate Fellowship via the College of Health and Human Development at The Pennsylvania State University (S Song). The funders had no role in study design, data collection and analysis, decision to publish, or preparation of the manuscript.

==============================
The present study examined the unique impact of perceived negativity in multiple social relationships on endocrine and inflammatory responses to a laboratory stressor. Via hierarchical cluster analysis, those who reported negative social exchanges across relationships with a romantic partner, family, and their closest friend had higher mean IL-6 across time and a greater increase in TNF-α from 15 min to 75 min post stress. Those who reported negative social exchanges across relationships with roommates, family, and their closest friend showed greater IL-6 responses to stress. Differences in mean IL-6 were accounted for by either depressed mood or hostility, whereas differences in the cytokine stress responses remained significant after controlling for those factors. Overall, this research provides preliminary evidence to suggest that having multiple negative relationships may exacerbate acute inflammatory responses to a laboratory stressor independent of hostility and depressed mood.

Stress is a routine part of daily life and interpersonal stress is often the most common and arguably the strongest type of stressor most people experience (Kiecolt-Glaser, Gouin & Hantsoo, 2010). Poorer overall health and dysregulated immune function are strongly linked with interpersonal stress both from negative social exchanges (Chiang et al., 2012; Edwards et al., 2001; Kiecolt-Glaser & Newton, 2001) and chronic social conflict (Cohen et al., 1998; Davis et al., 2008; Friedman et al., 2012). Interpersonal stress appears to have a long lasting impact on health in part by contributing to chronically elevated inflammation, which can confer risk of diverse age-related diseases (Ershler & Keller, 2000; Graham, Christian & Kiecolt-Glaser, 2007; Ridker et al., 2000). However, the majority of studies on immune responses to social conflict have focused on a particular type of relationship (e.g., marital relationships), while the effect of having conflict across multiple relationships is largely unknown. Further, the degree to which multiple social conflicts affect inflammatory responses to stress and whether the association is independent of related psychosocial characteristics are important issues that are not well understood. The present research is expected to advance the literature by examining interpersonal relationships in multiple areas and how negativity across multiple interpersonal relationships affects inflammatory responses to a laboratory stressor.

Studies most relevant to the current research have examined the effects of acute social conflict on health related outcomes. For example, the frequency of negative social exchanges with close others has been negatively associated with physical and mental health among college students (Edwards et al., 2001) and is predictive of depressed mood in a sample of married adults (Joiner & Timmons, 2009). Complementing such research, experimental studies have demonstrated that acute social conflict can influence immune responses in a laboratory setting (Chiang et al., 2012; Kiecolt-Glaser et al., 2005). One mechanism that may explain the negative effects of acute social conflict on health is repeated physiological activation of inflammatory stress responses and delayed recovery to stress (Seeman & McEwen, 1996). Under social conflict, inflammatory responses to stress may be also maintained by actions of the sympathetic-adrenal-medullary (SAM) system and the hypothalamic-pituitary-adrenal (HPA) axis (e.g., via cortisol) (Lovallo, 2005; Miller, Chen & Zhou, 2007). Recent studies show that gene expression of inflammatory pathways are upregulated in leukocytes among socially stressed individuals compared to matched control with good social support (Cole et al., 2011; Slavich & Cole, 2013).

In addition to a relatively direct effect of social conflict via stress activation, it is important to consider individual characteristics that tend to go along with negative social relationships. In particular, trait hostility and depressive symptoms appear to aggravate the effects of psychosocial stressors on cardiovascular and inflammatory response (Brondolo et al., 2003; Brummett et al., 2010). However, despite the possibility for hostility and depressed mood to be confounded with relationship stress and health, few relevant studies have controlled for hostility or depressed mood.

The present research aims to examine effects of negative social exchanges in multiple relationships—with a romantic partner, a close friend, close family members, and roommates—on responses to an experimental stressor, with an emphasis on inflammatory responses. We classified individuals into groups by their patterns of negative social exchanges across those four relationship areas. Because those who experience social conflict in many close relationships are a minority (Fingerman, Hay & Birditt, 2004), we expected to identify only a small number of those who reported negative social exchanges in multiple areas of measured relationships. We hypothesized that individuals with high levels of negative social exchanges in more relationships than others would have exaggerated or prolonged inflammatory response to stress (i.e., poorer recovery) in terms of two cytokines: IL-6 and TNF-α. TNF-α is a classic proinflammatory cytokine and IL-6, although it has anti-inflammatory actions in certain contexts (for reviews see Hawkley et al., 2007; Woods, Vieira & Keylock, 2009), is widely considered proinflammatory in the context of psychological stress. We examined whether the effects of negative relationships on inflammatory responses to stress are independent of depression or hostility. On a more exploratory basis, we also expected that individuals with more negative relationships would show greater increases in cortisol responses to the stressor.

Method

Participants

Fifty-six healthy participants (36 women, 20 men), aged 18–30 years (mean = 21.05 ± 0.37) were recruited to participate in a larger study examining influences of sex hormones to physiological responses to an experimental stressor. Participants were recruited via advertisements in the local newspaper and flyers posted in the local community and on the campus of a state university in the Northeastern U.S. An initial telephone interview was conducted by a trained research assistant to determine the eligibility of participants. Exclusion criteria included tobacco use, BMI ≥30, psychiatric hospitalization within the past year, the use of psychotropic medication, anti-inflammatory medications, hormonal contraceptives, medications for controlling blood pressure, and inhaled beta agonists. People who scored higher than the clinical cut off score of 16 on the Center for Epidemiologic Studies Depression Scale (Radloff, 1977) or with history of depression were not eligible for the larger study. In addition, we screened out people with a history of heart disease, diabetes, and neurological disorders. Women who reported any possibility for pregnancy and menstrual cycle dysregulation also were excluded. Women came to the laboratory during either the late luteal (n = 19) or follicular (n = 17) phase of their menstrual cycle for the purpose of the larger study.

Measures

Negative social relationships

A 25-item measure was used to assess negative social relationships, which was based on an existing questionnaire that included five items about negative social interactions with a spouse or significant other (Schuster, Kessler & Aseltine Jr, 1990). The items ask about the frequency of negative social exchanges involving disagreements, criticism, and tension, with responses ranging from 0 (never) to 5 (very often). The present study used those same five items to ask participants about negative social interactions among (a) roommates, (b) a romantic partner, (c) close family members, (d) their closest friend, and (e) their children. The alpha reliability of the original scale was 0.76 (Schuster, Kessler & Aseltine Jr, 1990), and the present sample showed Chronbach–α of 0.84, 0.89, 0.84, 0.81 for the romantic partner, family, roommate, and the closest friend subscales, respectively. As no participants reported having children, that subscale was not used.

Negative mood

A six item negative mood scale was administered four times (baseline, immediately after, 15 min, and 75 min after the stressor) to check the effect of the experimental stressor on mood. The scale consisted of words describing negative and positive mood (e.g., nervous, happy, irritated) with a 7-point Likert scale ranging from 1 (Not at all) to 7 (Very much). The scale showed a good internal consistency across measurements (Chronbach-α = 0.74, 0.88, 0.84, and 0.86, respectively).

Depressed mood

The Center for Epidemiological Studies Depression Scale (CES-D; Radloff, 1977) was used to measure depressed mood, which effectively identifies depressed mood among healthy individuals (Radloff, 1977). Item responses are from 0 to 3, with 3 representing the greatest frequency of depressed symptoms over the past week. The CES-D showed a chronbach-α of .90 for this sample.

Hostility

The well-validated Cook-Medley hostility questionnaire (CMHQ; Cook & Medley, 1954) was used to assess the tendency to react and think in a hostile manner. The scale has 50 true-false items, which are aggregated into a total score ranging from 0 to 50. The chronbach-α was .83 for this sample.

Procedures

Laboratory protocol and stressor

Eligible participants arrived at a General Clinical Research Center (GCRC) at 13:00 h and were met by a trained research assistant who first obtained a written informed consent. Next, participants were interviewed by a certified nurse practitioner to confirm health status and study eligibility. Participants then were asked to complete questionnaires, after which a trained nurse inserted an indwelling catheter in the non-dominant arm. After a 10 min acclimation period, participants were asked to sit quietly for 15 min. A baseline blood sample (20cc) was then drawn.

Next, a modified Trier Social Stress Task was administered. Participants were given 10 min to prepare a 3.5-minute speech about a personal failure that had a negative consequence on their life. They delivered the speech in front of a video camera and were told that a recording would be later observed by a panel of psychologists (no recording was actually made). Participants were prompted by the experimenter to continue talking if they finished their speech in less than the allotted time. Immediately after the speech, participants were asked to complete a serial subtraction task as fast and as accurately as possible (4 min), followed by several math word questions that increased in difficulty (3.5 min), and then another serial subtraction task (4 min). The experimenter delivered timed prompts to urge participants to work more quickly and to tell them to start over if they delivered the wrong response. This stress protocol took 30 min total.

Baseline blood samples and blood samples at 15 and 75 min after the stress period were used to determine cortisol, IL-6, and TNF-α. Participants completed several post-stress measures of mood at the end of the study, after which the catheter was removed. The study procedure was approved by the institutional review board at the Pennsylvania State University (IRB #00M0314-B9).

Blood handling

For preparation of serum, blood was drawn into separate collection tubes that contained no additive. Serum tubes were allowed to sit at room temperature for 15 min before centrifugation (1,500 × g at 4 °C for 15 min). Following centrifugation, serum was aliquoted into separate 100 µL microtubes and frozen at −80 °C for later assay.

Serum cortisol, IL-6, and TNF-α

Assays were performed at the Pennsylvania State University GCRC Core Laboratory. Serum cortisol levels were determined using commercially available enzyme immunoassay kits (EIA; Diagnostic Systems Laboratories, Inc., Webster, TX). The inter-assay and intra-assay coefficients of variation were 3.16% and 4.8%, respectively for cortisol. Serum IL-6 and TNF-α levels were determined by enzyme-linked immunosorbant assays constructed with antibodies purchased from R&D systems (Minneapolis, MN) using previously described procedure (Corwin et al., 2003). The level of detection was 1.0–3.0 pg/mL, and inter-assay and intra-assay coefficients of variation were 7.1% and 5.3% respectively for these cytokine assays. All samples were tested in duplicate in a single assay batch; values that varied by more than 5% were subject to repeat testing. The average of duplicate tests is reported for each biomarker assay.

Data analysis

SPSS 20.0 was used for all analyses. Study variables were screened for outliers and non-normality, and cortisol, IL-6, and, TNF-α were natural log transformed to correct for skewness. Next, a hierarchical cluster analysis was applied to classify individuals with different patterns of perceived negativity across the relationships with roommates, a romantic partner, close family, and a closest friend. Out of 56 participants, one woman did not provide sufficient data to compute the cluster by negative relationship analyses and was therefore excluded from analyses. While 25 participants reported having all of the four relationships, 13 participants did not have a roommate, and 13 others did not have a romantic partner. Thus, two separate hierarchical cluster analyses were run, the first cluster analysis including those who reported having a romantic partner, a closest friend, and family (n = 38) and the second including individuals who reported having roommates, a closest friend, and family (n = 38). Four participants who did not report a relationship with either a romantic partner or roommates were excluded from these cluster analyses; there was no significant difference between those four participants and the rest of the sample in any psychological characteristics or outcome variable we examined.

In each of two cluster analyses conducted, the same three steps were used. First, the variables for the perceived negativity in the three relationship areas were entered via hierarchical cluster analysis. Ward’s method with the similarity measure of squared Euclidean distance was then used to decide the number of groups in the cluster model. Discriminant function analysis was used to verify how much of the clustering within groups could be replicated (Klecka, 1980). In the third step, we further examined the characteristics of the groups in the cluster model via F tests by examining whether the groups differed by age, gender, and the four negative relationship variables.

The general linear model (GLM) with within-subject design was then used to examine the effect of the different negative social relationship clusters on cortisol and cytokine responses to stress. Greenhouse-Geisser correction was used for sphericity. For the significant results, the partial eta-squared (ηp2) post hoc tests with Bonferroni correction were reported. As post hoc tests for the time effects, difference scores were calculated for the stress response measures between each pair of the three time points (e.g., from baseline to 15 min after stress) in order to examine change during each time interval. Due to their known impact on inflammation, age, BMI, gender, and menstrual cycle of women were controlled in all of these analyses. Three dummy coded variables representing (a) men, (b) women in the luteal period, and (c) women in the follicular period were generated and entered in analyses to control for both gender and women’s menstrual cycle status. Finally, depressed mood and hostility were additionally entered to the GLM to examine whether the effects were independent of those characteristics.

Results

Preliminary analyses

The means and standard deviations (SDs) of demographics and study variables are presented in Table 1. The sample was predominantly Caucasian (71%) and comprised of young adults with a mean age of 21.05 (SD = 2.74, range 18–30).

Table 1 Sample characteristics and the baseline measure of biomarkers by cluster groups.

			Total (N = 56)	
			M	SD	
Characteristics	
Age (yrs)	21.05	2.74	
Women (%)	64.3		
Cycling status among women (n = 36)			
Luteal (%)	52.78		
Follicular (%)	47.22		
Body Mass Index (kg/m2)	23.42	3.10	
Hostility	23.32	6.85	
Depressed mood	9.29	8.30	
Baseline measure of biomarkers by cluster groups			
Biomarker	Model	Groups	M	SD	
Cortisol (μ g/dL)	Romantic partner model	Low conflict	2.38	0.45	
		Multiple conflict	2.23	0.43	
	Roommate model	Low conflict	2.38	0.42	
		Only family conflict	2.02	0.37	
		Multiple conflict	2.47	0.20	
		Total	2.37	0.42	
IL-6 (pg/mL)	Romantic partner model	Low conflict	2.96	0.97	
		Multiple conflict	3.74	1.15	
	Roommate model	Low conflict	3.41	1.40	
		Only family conflict	4.25	0.78	
		Multiple conflict	3.95	1.10	
		Total	3.46	1.23	
TNF-α (pg/mL)	Romantic partner model	Low conflict	3.07	0.93	
		Multiple conflict	3.31	1.08	
	Roommate model	Low conflict	3.44	1.21	
		Only family conflict	4.18	1.27	
		Multiple conflict	3.79	0.78	
		Total	3.42	1.11	
Notes.

Biomarker levels were log transformedμg/dL micrograms per deciliter

pg/mL picograms per milliliter

Cluster analysis for negative relationship profiles

We classified individuals as having different degrees of negative social exchanges across multiple interpersonal relationship areas. The first cluster analysis was run on the 38 participants who had a romantic partner, family, and a close friend. It yielded two groups (Table 2A): “a low conflict group” (n = 29) characterized by consistently low levels of negative social exchanges across all relationship areas (romantic partner, family, and their closest friend), and “a multiple conflict group” (n = 9) that had high levels of negative social exchanges across all relationship areas. F tests confirmed that the two groups in this cluster model were different in levels of negative social exchanges in these relationships (ps < .05). A discriminant function analysis verified the cluster structure (χ2 (3, n = 38) = 53.56, p < .001), and 97.4% of the original grouped cases (37 cases out of 38) were replicated. The distribution of gender and age was not significantly different across the two groups.

Table 2 The level of negative social exchanges in each relationship area for the cluster groups, generated (A) by relationships with a romantic partner, family, and the closest friend and (B) by relationships with roommates, family, and the closest friend.

(A)	
Negative social exchanges in the relationships with	The Romantic Partner Model	
	Low conflict (n = 29)	Multiple conflict (n = 9)	
	M	SD	M	SD	
Romantic partner	7.86	2.45	11.89	4.91	
Family	8.07	2.15	16.44	1.88	
The closest friend	6.72	2.10	8.89	3.92	
(B)	
Negative social exchanges in the relationships with	The Roommates Model	
	Low conflict (n = 30)	Only family conflict (n = 5)	Multiple conflict (n = 3)	
	M	SD	M	SD	M	SD	
Roommates	8.60	3.40	6.60	2.30	17.00	3.46	
Family	8.60	2.75	17.60	1.95	20.33	2.52	
The closest friend	6.53	1.89	7.80	3.03	16.00	1.73	

Another cluster analysis was run on the 38 participants who reported relationships with roommates, family, and the closest friend. This analysis identified three clusters (Table 2B): “a low conflict group” (n = 30) characterized by low levels of negative social exchanges across all the relationship areas, “a multiple conflict group” (n = 3) characterized by high levels of negative social exchanges across all the relationship areas, and “a family conflict group” (n = 5) characterized primarily by a high level of negative social exchanges in family but low levels of negative social exchanges among roommates and the closest friend. The F tests confirmed that the three groups in this cluster analysis (hereafter referred to as the “roommate model”) were different in levels of negative social exchanges across the three relationship areas (ps < .001). A discriminant function analysis verified the three cluster structure (χ2 (6, n = 38) = 70.69, p < .001) and that 94.7% (36 cases out of 38) of the original grouped cases were replicated. The gender distribution was significantly different across the 3 groups (p = .05), which was largely driven by the multiple conflict group having only three men and no women. Age was not different across the 3 groups.

Manipulation checks for the stress protocols

Participants’ negative mood increased in response to the stressor (F(3, 153) = 34.71, p < .001, ηp2 = .28). The levels of serum cortisol did not significantly increase in response to the experimental stressor but showed a significant decrease over time (F(2, 106) = 9.88, p < .001, ηp2 = .16), likely driven primarily by the diurnal rhythm of cortisol. There was no significant time effect on IL-6 and TNF-α levels.

Baseline differences in biomarkers by negative social relationships

The Table 1 presents the baseline levels of biomarkers by cluster groups in both the romantic partner model and the roommate model. There was a significant baseline difference in IL-6 levels between the low and multiple conflict groups in romantic partner model (F(1, 36) = 4.18, p = .05, ηp2 = .10). The baseline difference became not significant after controlling for depression or hostility along with age, gender, and menstrual cycle status. There were no other baseline differences in any of the groups in either the romantic partner or roommate model.

Stress responses by negative social relationships

IL-6

There was a significant time by negative social relationship interaction using the roommates model on IL-6 (F(4, 58) = 8.53, p < .01, ηp2 = .37). Post hoc tests confirmed that only individuals in the multiple conflict group of this cluster model showed significantly greater increases in IL-6 from baseline to 15 min after stress (ps < .01) and from baseline to 75 min after stress (ps < .01) compared to those in the family conflict or low conflict groups (Fig. 1). Results remained significant after controlling for depressed mood or hostility (ps < .01).

Figure 1 Changes in serum IL-6 levels by the negative relationship groups in the roommate model.

LN, Natural Log transformation.

TNF-α

There was a marginally significant time by negative social relationship interaction among the romantic partner model on TNF-α responses to the stressor (F(2, 56) = 2.80, p = .07, ηp2 = .09). Upon examination, individuals in the multiple conflict group (who reported negativity in their relationships with their romantic partner as well as their closest friend, and family) showed significantly greater increases in TNF-α from 15 min to 75 min post stress after controlling for baseline TNF-α and other covariates (F(1, 27) = 6.81, p < .05, ηp2 = .20), as illustrated in Fig. 2. This remained significant after controlling for depressed mood or hostility (ps < .05) and also after removing the one overlapping individual in the multiple conflict group who was also included in the multiple conflict group in the roommate model (where greater IL-6 responses to stress were observed). There was no main effect in the roommate model groups on TNF-α stress response.

Figure 2 Changes in serum TNF-α levels by the negative relationship groups in the romantic partner model.

LN, Natural Log transformation.

Cortisol

Neither of the negative relationship cluster models significantly predicted cortisol responses to the stressor.

Discussion

Although social conflict has been associated consistently with poorer health and various stress-related biomarkers (Graham, Christian & Kiecolt-Glaser, 2007; Kiecolt-Glaser & Newton, 2001), the majority of past studies showing connections between relationship stress and biomarkers have focused on a particular type of relationship or broad characterizations of relationship quality or network size. The present research is the first to examine the effect of negative social exchanges across multiple interpersonal relationships and whether the effects of negative social exchanges in multiple relationship areas are independent of depressed mood and trait hostility. Another novel aspect of the present research is that we focused on the effect of relationships on inflammatory responses to stress as opposed to basal levels of biomarkers. As expected, the present results suggest that there are differences in acute inflammatory cytokine responses to stress depending on the pattern of multiple negative social relationships individuals reported within the four relationship areas examined (romantic partner, family, the closest friend, and roommates). Those who reported negative social exchanges in their relationships with roommates, family, and their closest friend showed increases in IL-6 after being exposed to a laboratory stressor. Similarly, people who reported negative social exchanges with a romantic partner, family, and their closest friend showed increases in TNF-α from 15 min to 75 min post stress after controlling for baseline TNF-α. Both the IL-6 and the TNF-α results remained significant after controlling for depressed mood, hostility, age, BMI, gender, and menstrual cycle status.

Importantly, the two multiple conflict groups examined in this research (among those who had roommates, vs. those who had a romantic partner) were largely distinct from each other; there was only one individual who was included in both of these analyses. Thus, we showed an effect of negativity across multiple relationship areas on increases in either TNF-α or IL-6 via different subsets of individuals, suggesting that the effects of having social conflict across relationship areas are robust and pro-inflammatory in nature. Further, results were not driven by baseline effects in inflammation or outliers. We did not find any outliers in either of the groups that showed significant stress responses and in all participants. Further, as compared to the other groups, individuals in the multiple conflict group of the roommate model did not evidence significantly different baseline levels of IL-6 and those in the multiple conflict group of the romantic partner model did not evidence significantly different baseline levels of TNF-α. Thus, participants in those two multiple conflict groups came to the lab without elevated IL-6 or TNF-α compared to others, but were the ones who showed increases in IL-6 or TNF-α after being exposed to the laboratory stressor.

The findings of the present research complement the results of a recent study showing that daily levels of negative social interactions were associated with greater inflammatory responses to an experimental stressor as measured by IL-6 and soluble TNF-α receptor II (Chiang et al., 2012). In terms of direction, the inflammatory stress reactivity of individuals with multiple negative relationships is also consistent with previous studies of the association between a laboratory induced social conflict and inflammatory responses (Graham, Christian & Kiecolt-Glaser, 2007; Kiecolt-Glaser et al., 2005). Psychological stress effects on excessive inflammatory cytokines responses are likely explained by multiple aspects of complex, interrelated physiological systems. For example, chronic interpersonal stress is likely be related to dysregulation of the inflammatory stress response due to decreased glucocorticoid receptor sensitivity (Corwin et al., 2013; Miller, Chen & Zhou, 2007; Pace et al., 2012) or perhaps to down-regulation of cholinergic anti-inflammatory pathways in neural circuitry (Tracey, 2002).

Interestingly, there was one baseline difference in IL-6 observed: Participants who reported multiple conflict across relationships and who had a romantic partner had higher IL-6 at baseline, an effect that was reduced to non-significance when controlling for either depressed mood or hostility and which was not observed among those who reported multiple relationship conflict and who had roommates. It may be that for this relatively young sample, which was largely comprised of college students (78%), that those who were in conflictive romantic relationships were more depressed or hostile and that this (rather than the conflict itself) explained their higher baseline IL-6. In contrast, those in the present study who had conflicts with roommates may have had less control over their exposure to those individuals and may have developed greater inflammatory stress reactivity due to frequent exposure to stress. The difference observed between those with roommates and romantic partners was not expected, and may be limited to the particular sample in the present research. However, the finding that depressed mood and hostility can explain baseline levels of inflammation is consistent with previous research showing that depressed mood and hostility are associated with increased circulating markers of inflammation among adults (Graham, Christian & Kiecolt-Glaser, 2006; Suarez, Lewis & Kuhn, 2002; Zorrilla et al., 2001). Importantly, no such baseline effects explained the effects of relationship negativity on the inflammatory cytokine responses to acute stress observed in the present research.

We did not find a significant effect of negative relationship endorsement on cortisol responses. This null effect might be related to the experimental stress paradigm used in which the public speech part of the stress task was conducted in front of a video camera instead of a panel of judges, a protocol which can reduce the intensity of stress response from the Trier (Dickerson & Kemeny, 2004; Kirschbaum, Pirke & Hellhammer, 1993). Also, the timing of the catheter insertion and blood draws might also explain the null finding.

Limitations

The clinical implications of the study are limited in several ways. The present research was conducted with a small sample of healthy, relatively young adults, which limits generalizability and warrants cautions in interpretation. However, a relatively small proportion of a sample can be expected to have multiple relationship conflicts and the issues with there being a small number of those with multiple conflicts were minimized in the present research by our application of conservative statistical adjustments when using cluster analysis and general linear model and comparisons of analyses between the two multiple conflict clusters (Clatworthy et al., 2005; Hair et al., 2006; Huynh, 1978). It will be important to replicate the present results using larger and more diverse samples, particularly a greater number of individuals reporting negative relationships across multiple social relationship areas than was available in the present study. It was also a limitation that we only had IL-6 and TNF-α available as inflammatory biomarkers: Future research on related topics would benefit from utilizing multiplex technology to examine a greater diversity of analytes, including anti-inflammatory cytokines (e.g., IL-10) to enable examination of the ratio between pro and anti-inflammatory cytokines in stress responses. It would also have been ideal if we had been able to include blood draws later than 75 min post-stress to capture the full peak as well as return to baseline of the cytokine responses. Finally, it would be of value for future research to utilize clinically diverse samples, such as those with clinical depression or existing inflammatory conditions, and to include assessments of clinical health outcomes.

Conclusion

A better appreciation for how social conflict may alter stress responsiveness of the body is critical to understanding how and why it is associated with poorer physical health (Seeman & McEwen, 1996). Although having some degree of social conflict is an unavoidable part of everyday life, the present study provides preliminary evidence that having negative social interactions across multiple social relationships might be harmful, as it is associated with greater inflammatory reactivity to a psychosocial stressor. Significant increases in both IL-6 and TNF-α in response to stress were observed among those with relationship conflict in at least three areas, compared to those with relationship conflict in fewer relationships. The effect of having multiple social relationships on inflammatory responses to stress appears to be independent of any effect of hostility or depressed mood. Taken as a whole, the present research emphasizes the importance of examining the role of negative close relationships in inflammatory stress response in a detailed fashion. Having multiple negative relationships may put individuals at particular risk of developing exacerbated acute inflammatory reactivity to psychosocial stress.

Supplemental Information

Supplemental Information 1 Song et al. Dataset

Click here for additional data file.

We appreciate the dedicated assistance of undergraduate students in the Biobehavioral Health Studies Lab, the GCRC nursing staff, MM Stine and CA Whetzel.

Additional Information and Declarations

Competing Interests

Author Contributions

Human Ethics

The authors declare there are no competing interests.

Sunmi Song analyzed the data, wrote the paper, prepared figures and/or tables, reviewed drafts of the paper.

Jennifer E. Graham-Engeland contributed reagents/materials/analysis tools, reviewed drafts of the paper.

Elizabeth J. Corwin and Laura Cousino Klein conceived and designed the experiments, performed the experiments, contributed reagents/materials/analysis tools, reviewed drafts of the paper.

Rachel M. Ceballos conceived and designed the experiments, performed the experiments, reviewed drafts of the paper.

Shelley E. Taylor conceived and designed the experiments, contributed reagents/materials/analysis tools, reviewed drafts of the paper.

Teresa Seeman conceived and designed the experiments, reviewed drafts of the paper.

The following information was supplied relating to ethical approvals (i.e., approving body and any reference numbers):

The Pennsylvania State University Institutional Review Board: IRB # 00M0314-B9.

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
