# Peer review of "The role of multiple negative social relationships in inflammatory cytokine responses to a laboratory stressor"

_PeerJ, doi:10.7717/peerj.959_

## Round 0.1 · original submission · Minor Revisions

Please revise the manuscript to incorporate the useful and constructive suggestions of the reviewers.

Reviewer 1 ·

Basic reporting

I think it would be useful to the reader if you provided a table of baseline values of IL 6 and TNF in the different groups. Table 1 shows only the total and is uninformative due to the enormous SD of the mean for the cytokines, unlike the cortisol.

Fig. 1 shows only IL 6. Please include data on TNF also in graphical form.

Experimental design

I cannot comment on the pyschosocial measures and statistics.

Please provide details of the cytokine assay measurements. You refer to a paper from 1999 by one of the authors, but this assay is not the "gold-standard" nowadays. In any case, that paper is on other cytokines and published in a hard-to-access journal. You have measured duplicates and averaged them, but how many times did you repeat the assay? What is its sensitivity and reproducibility?

You show increases in IL 6 after the stressor at 15 min and maintained at 75 min, but you say that baseline was no different between groups. It would therefore be important to assess "resolution" of the acute response at times later than 75 min.

Validity of the findings

TNF and IL 6 are usually considered pro-inflammatory, but as the authors recognize, at least for IL 6 this is an oversimplification. The title of the paper is thus not completely accurate. Apart from that, focussing on only two cytokines when modern analytical techniques allow multiplexing from similarly small amounts of serum greatly limits the study. It might be particularly relevant to measure "anti-inflammatory" cytokine as well, especially IL 10. It may be the ratio of pro- and anti-inflammatory cytokines that is important. This could a reason for the puzzling differences in associations between TNF and IL 6 with the various different negative social relationships observed.

As the authors also point out, this a preliminary study and although their statistical analysis indicates validity of the cohort size (I am unable to comment on this), it does strike me as a priori very small to measure effects of such a variety of factors.

Additional comments

An intriguing study but hard to interpret biologically, as you only looked at two serum cytokines with different properties and not only pro-inflammatory. In the mostly negative data, where you did see an acute response, it would important to determine response resolution, as the baseline values in the different groups were the same. As you point out, the study population is very small and it seems to me that these preliminary data may be too preliminary.

·

Basic reporting

No comments

Experimental design

No comments

Validity of the findings

No comments

Additional comments

This study contributes to the field trying to delineate the relationships between stressful social interactions, personality characteristics and health outcomes (both physical and mental). Although there are limitations to the sample investigated (recognized by the authors), being relatively healthy, young people, it provides evidence for the ability to delineate factors that relate to baseline immune characteristics, versus reactive immune characteristics. It is of interest that subjects who appeared immunologically similar at baseline, demonstrated differences in dynamic response to stress, differences in part explained by their degree of social conflict. It would indeed, as the authors indicate, be interesting to see such a study design repeated with a broader range of participants and clinical statuses. Overall, this study presents novel data useful to informing future directions for the field.

A clarification would be useful:

1) In the results section the authors examine the difference between mean IL-6 for the groups in the romantic partner model. Given that Figure 1 demonstrates no difference in baseline measures of IL-6 (even if this graph shows the room-mate model rather than the 2 group model), would it not be more clear to indicate that the difference between the two groups reflects a difference in stress response rather than baseline, and therefore to focus on the post-stress levels of IL-6. Otherwise please justify the use of mean IL-6 as an outcome of interest.

---

## Round 0.2 · accepted · Accept

All the concerns raised have been convincingly addressed.